# Deep Learning-Based Ground-Penetrating Radar Inversion for Tree Roots in Heterogeneous Soil

**DOI:** 10.3390/s25030947

**Published:** 2025-02-05

**Authors:** Xibei Li, Xi Cheng, Yunjie Zhao, Binbin Xiang, Taihong Zhang

**Affiliations:** 1School of Computer and Information Engineering, Xinjiang Agricultural University, Urumqi 830052, China; 320223340@xjau.edu.cn (X.L.);; 2Xinjiang Agricultural Information Engineering Technology Research Center, Xinjiang Agricultural University, Urumqi 830052, China; 3School of Mechanical Engineering, Xinjiang University, Urumqi 830017, China

**Keywords:** ground-penetrating radar, tree root detection, layered heterogeneous soil, permittivity inversion, deep learning

## Abstract

Tree roots are vital for tree ecosystems; accurate root detection helps analyze the health of trees and supports the effective management of resources such as fertilizers, water and pesticides. In this paper, a deep learning-based ground-penetrating radar (GPR) inversion method is proposed to simultaneously image the spatial distribution of permittivity for subsurface tree roots and layered heterogeneous soils in real time. Additionally, a GPR simulation data set and a measured data set are built in this study, which were used to train inversion models and validate the effectiveness of GPR inversion methods.The introduced GPR inversion model is a pyramid convolutional network with vision transformer and edge inversion auxiliary task (PyViTENet), which combines pyramidal convolution and vision transformer to improve the diversity and accuracy of data feature extraction. Furthermore, by adding the task of edge inversion of the permittivity distribution of underground materials, the model focuses more on the details of heterogeneous structures. The experimental results show that, for the case of buried scatterers in layered heterogeneous soil, the PyViTENet performs better than other deep learning methods on the simulation data set. It can more accurately invert the permittivity of scatterers and the soil stratification. The most notable advantage of PyViTENet is that it can accurately capture the heterogeneous structural details of the soil within the layer since the soil around the tree roots in the real scene is layered soil and each layer of soil is also heterogeneous due to factors such as humidity, proportion of different soil particles, etc.In order to further verify the effectiveness of the proposed inversion method, this study applied the PyViTENet to GPR measured data through transfer learning for reconstructing the permittivity, shape, and position information of scatterers in the actual scene. The proposed model shows good generalization ability and accuracy, and provides a basis for non-destructive detection of underground scatterers and their surrounding medium.

## 1. Introduction

Tree roots play a crucial role in plant ecosystems. Therefore, root system detection is important both in the agricultural field and from the perspective of plant ecological protection.Accurate detection of tree roots helps analyze and evaluate tree health risks and achieve reasonable management and control of fertilizer, water, and medicine in tree planting. Root detection methods are mainly divided into two categories: destructive detection and non-destructive detection. Traditional destructive detection methods include root drilling [1], soil core method [2], soil profile method [3], and full root excavation method [4].These methods can be time-consuming, labor-intensive, and cannot be applied on a large scale. The most critical thing is that the detection process of these methods may cause irreversible damage to the tree root system.

Ground-penetrating radar (GPR) is a non-destructive detection technology that is widely used to detect and image underground structures, strata, and features of buried objects. By emitting high-frequency electromagnetic pulses and receiving signals reflected from the interfaces of different materials, GPR can detect tree roots without destroying the underground medium.

Traditionally, the analysis of GPR data mainly relies on the skills and experience of professionals, which makes it subjective and not real-time analysis. Therefore, GPR data inversion methods are urgently needed, which are imaging technology that uses the quantitative information of the echo received by the receiver to infer the position, shape and the electrical properties (such as permittivity and conductivity) of the target medium or scatterers [5,6,7,8,9].

There are two major categories of GPR inversion methods: classical methods and deep learning-driven methods. The traditional GPR inversion methods for reconstructing underground medium structure from echo data include migration algorithms, microwave tomographic imaging and full waveform inversion (FWI) [10]. The migration algorithm converts the unfocused GPR image into a focused image, thereby displaying the position and size of the target scatterer or medium interface. However, the migration algorithm cannot provide the electrical parameters of the scatterer, which is not conducive to the identification and status monitoring of underground media. Microwave tomographic imaging approach can reconstruct the electrical parameters of shallow surface media using GPR data, but it usually has high measurement configuration requirements and needs to meet certain restrictions before it can be used. This method is mostly used for borehole radar inversion problems. At the same time, the large number of iterative inversion processes is computationally expensive. The above two methods only use part of the data information in the electromagnetic echo. In contrast, FWI uses all the waveform information in the echo signal. The application of the FWI method originated in the field of time-domain seismic wave imaging [10]. Electromagnetic waves and seismic waves have high similarities in dynamic and kinematic characteristics, which provides a basis for the extension of the FWI method from the field of seismic exploration to the field of GPR research.

It achieves imaging by matching the numerical simulation model of underground materials with the actual collected radar data through an iterative optimization process. since the FWI method need to perform forward simulations multiple times in each inversion iteration, which places extremely high demands on computing resources. It is computationally intensive and requires a pre-estimated permittivity distribution of underground object or medium as the initial value. Additionally, it is easily affected by the accuracy of the initial value and the quality of the data, and is prone to falling into local minima.

With the rapid development of deep learning technology, a series of innovative GPR inversion algorithms have emerged [11]. One of the deep learning models applied for GPR inversion earlier is U-Net, which achieves the inversion of electrical parameters of objects in homogeneous soil, but the inversion task of U-Net is only for scatterers buried in the soil, and it cannot accurately identify the structure of the background medium [12]. The GPRInvNet model focuses on solving the problem of dielectric property inversion of underground tunnel lining structures. It takes GPR B-scan images as input and outputs the relative permittivity distribution of underground pipelines, which can accurately depict the contours of complex tunnel defects [13]. The DMRF-UNet model exhibits excellent adaptability to the complexity of heterogeneous soils. It was tested in a variety of heterogeneous soil environments and successfully reconstructed the dielectric constant, morphology, size, and location of underground targets. Through a two-stage training strategy and multi-scale convolution techniques, DMRF-UNet achieves highly accurate predictions. However, the complexity of the model training process, the huge number of model parameters and the slow inversion speed provide directions for further optimization of the DMRF-UNet model [14]. The TransUNet model consists of a residual convolution encoder, a self-attention encoder and a decoder. It works well when dealing with data of layered soils where the soil in each layer is homogeneous, but when it was applied to more complex intra-layer heterogeneous soil, the current method has limitations in recognition of more subtle morphological changes within each layer of heterogeneous soil [15]. The EDMFEBs model proposed greedy channel-spatial attention and multi-scale feature extraction modules, which has good performance in inverting the electrical parameters of scatterers in heterogeneous medium, but its accuracy in inverting the electrical parameters of heterogeneous medium needs to be improved [16].

In practical applications, not only the electrical parameters of buried objects are required, but also information about the background medium around the objects is needed. For example, the permittivity of the soil around tree roots is related to soil moisture, which is important for irrigation [17]. Moreover, the soil around tree roots is usually layered heterogeneous soil, and the soil within the layers is also heterogeneous due to factors such as soil compaction and humidity. In this work, a pyramid convolutional network with vision transformer and edge inversion auxiliary task called PyViTENet is proposed to address the challenges of electrical parameter inversion of underground tree roots and heterogeneous soil around them. The PyViTENet inversion model can accurately and simultaneously image the underground objects and the layered heterogeneous soil around the objects. Also, details of the heterogeneous structure within the layered heterogeneous soil are identified from the GPR data by the proposed method. The PyViTENet inversion model improves the data feature extraction ability by introducing pyramidal convolution and vision transformer, and increases the model’s attention to the details of the heterogeneous structure by adding the task of inverting the edge of the permittivity distribution of underground materials. Moreover, a GPR simulation dataset and a measured dataset are constructed in this study, respectively, and they are applied to verify the effectiveness and generalization ability of the proposed method.

This paper is organized as follows. Section 2 gives the details of the PyViTENet inversion model. The experiment results and analysis of the GPR simulation data are given in Section 3, and the experiment results and analysis of GPR measurement data are presented in Section 4. Conclusions and perspectives are given in Section 5.

## 2. Materials and Methods

### 2.1. Overall Architecture

GPR data inversion is an imaging technology that quantitatively estimates the location, structure, and electrical parameters of the target medium or scatterer through the received underground reflection echo information. Figure 1 shows the architecture of the proposed GPR data inversion model PyViTENet. It uses GPR B-scans as inputs, and outputs the permittivity map of underground objects and the layered heterogeneous soil around the objects. In addition to the main inversion task, PyViTENet contains an auxiliary inversion task which focuses on inverting the edge of the permittivity distribution map. This auxiliary task is used to enhance the model’s ability to identify heterogeneous structures inside the soil. The main inversion task consists of three parts: encoder, feature extractor, and decoder. The encoder uses a convolutional layer with a 3×3 convolution kernel, a batch normalization (BN) layer [18], and the rectified linear unit (ReLU) [19] activation function to encode the input features. The feature extractor consists of four pyramidal convolution feature extraction blocks (PyConvFEBs) and one vision transformer feature extraction block (ViTFEB). The decoder is constructed by a convolutional layer with a 3×3 convolution kernel followed by a Sigmoid activation function, which maps the result to between 0 and 1, and merges the channels into 1 channel as the final output of the task.

The auxiliary inversion task part also consists of three parts: encoder, feature extractor, and decoder. The encoder and decoder have the same structure as in the main inversion task part. The output channels are 64 and 1 respectively. The feature extraction part consists of five PyConvFEBs. The feature maps of the first, second, fourth, and fifth PyConvFEBs are channel connected to the corresponding main inversion task part in order to give the main inversion task more information about the edge of the permittivity distribution map. The output channels of the five modules are 64, 128, 128, 64, and 32 in sequence. Since the complexity of the edge inversion task is lower than that of the main inversion task, and to reduce the GPU memory required for model training and the computational burden, ViTFEB is not added to the auxiliary inversion task, but PyConvFEB is used instead.

### 2.2. Pyramidal Convolution Feature Extraction Block

Since the vertical axes of the GPR B-scan image and the permittivity distribution map of underground materials represent time and depth, respectively, the two cannot directly correspond. It requires the inversion model to extract regular information from a larger-scale receptive field. Therefore, this paper defines a method called PyConvFEB that uses pyramidal convolution (PyConv) [20] for multi-scale feature extraction. The structure of PyConvFEB is shown in Figure 2. In the figure, *k* represents the convolution kernel size, *s* represents the convolution step size, *p* represents the number of paddings, and *g* represents the number of groups. PyConv can process input data on multiple scales of convolution kernels. It contains a kernel pyramid, where each level contains different types of convolution kernels with different sizes and depths. These convolution kernels discover different levels of details in the input feature map and can improve the recognition ability of the model [20]. The information extracted by different types of convolution kernels is complementary. Kernels with smaller receptive fields can focus on details and capture information about smaller objects and detailed structures within larger objects, while larger receptive fields can extract contextual information and discover the rules of the non-corresponding parts of the spatial position between the input feature map and the output feature map. PyConv can effectively capture more detailed and nuanced information by interpreting input data across multiple scales while maintaining computational cost [20]. The PyConvFEB module contains three parallel convolution operations, which are convolution with a 3×3 kernel with the number of convolution groups being 1, convolution with a 5×5 kernel with the number of convolution groups being 4, and convolution with a 7×7 kernel with the number of convolution groups being 8. This structure can extract image features of different scales and different levels of detail through multi-scale convolution kernels and different group convolution configurations.

Then, to preliminarily obtain the feature map, the output features of different scales from the three parallel convolutions are concatenated together, and the number of feature channels is adjusted to the number of target output channels through a 3×3 convolution layer followed by the BN and ReLU activation function. At the end of the module, the convolutional block attention module (CBAM) [21] is introduced to input the preliminary feature map and add the output to the original value to further enhance the feature representation.

### 2.3. Vision Transformer Feature Extraction Block

The ViTFEB module consists of serial multi-scale feature extraction operations and two vision transformer blocks (ViTBlocks), and the structure of ViTFEB is shown in Figure 3. In the figure, *k* represents the convolution kernel size, *s* represents the convolution step size, and *p* represents the number of paddings. ViTFEB performs initial feature extraction on the input feature map through a 7×7 convolution kernel, and uses two 3×3 kernel convolutions in subsequent layers to refine features. Then, a 5×5 kernel convolution and three 3×3 convolutional layers are used to further extract features. The above serial multi-scale feature extraction converts the 256-channel 128×128 feature map into the 256-channel 32×32 feature map, which reduces the feature map size and improves the subsequent calculation efficiency of ViTBlocks. The ViTBlock consists of two Transformer encoders [22]. The core modules of the Transformer encoder are multihead self-attention (MSA) and multilayer perceptron (MLP), of which the number of MSA heads is 16, and the MLP mapping dimension is 512. LayerNorm is normalized on the feature dimension of each sample. The input feature map is divided into a series of 1×1 patches, and these patches are converted into fixed-dimensional vectors. These vectors are then fed into ViT for processing. These layers capture the global context information while processing the input sequence [22]. The calculated results are added to the original feature map, and then a 6×6 deconvolution, three 3×3 convolutions, and a 4×4 deconvolution are used to restore the feature map size, and the feature representation is further enhanced by the CBAM module.

## 3. Experiments and Analysis of Synthetic Data

### 3.1. Building the Synthetic Dataset

#### 3.1.1. Overview of Simulation Model

Figure 4 presents the GPR model used for simulations in this study. The transmitter TX emits electromagnetic signals to the underground, and the receiver RX receives the electromagnetic waves reflected from the underground. The TX and RX move in the direction of the arrow to generate multiple channels of GPR A-scan data, which are one-dimensional data. These GPR A-scan data are merged into B-scan data, which are two-dimensional data. An open source simulation software gprMax [23] is applied to build the above GPR model and generate simulation data in this study. gprMax uses the Finite-Difference Time-Domain (FDTD) [23] method to solve Maxwell’s equations in three dimensions, which can accurately simulate electromagnetic wave propagation in complex media environments. The type of antenna design used by default in gprMax is the bow-tie antenna.

In this study, the excitation in gprMax for GPR simulations is a Ricker waveform with an amplitude of 1 A, a center frequency of 900 MHz, and the time window is set to 20 ns [24]. The transmitter TX and the receiver RX are located at a height of 0.005 m from the ground, and the horizontal spacing between them is fixed at 0.14 m. At the beginning of the experiment, the transmitter TX moves to the right and transmits the signal from the position of 0.025 m. Currently, the receiver RX is at the position of 0.165 m. The transmitter TX and the receiver RX slide synchronously. When the transmitter moves to the rightmost position of 1.375 m, the receiver reaches the position of 1.515 m accordingly. During the whole sliding process, the electromagnetic signal is emitted once every 0.015 m, and a total of 90 channels of data are collected.

The permittivity distribution maps are chosen as the training label for the dataset instead of the conductivity distribution maps. The permittivity is directly related to the propagation time and reflection intensity of the electromagnetic waves, both of which can be extracted from GPR data with high accuracy. This makes it particularly suitable for detailed soil property inversion. In contrast, Conductivity primarily influences signal attenuation, which tends to have lower resolution and is affected by multiple factors, such as noise, antenna characteristics, and varying environmental conditions. As a result, extracting precise conductivity distributions from B-scan data remains more difficult.

#### 3.1.2. Layered Heterogeneous Soil and Buried Objects

The soil structure in the GPR model is divided into three layers based on the root growth characteristics of walnut trees in Kashgar, Xinjiang and the specific conditions of the surrounding soil [25]. Each layer is designed according to a specific humidity range and temperature conditions, thus forming a layered heterogeneous soil model. The details are as follows: the volume moisture content of the first layer of soil is 5∼10% and the temperature is 20 °C; the volume moisture content of the second layer of soil is 10∼13% and the temperature is 17 °C; and the volume moisture content of the third layer of soil is 13∼15% and the temperature is 15 °C.

In addition to temperature and humidity, different soil textures are also selected for each layer, aiming to increase the diversity of the data. Soil texture is one of the inherent basic properties of soil. It reflects the characteristics of soil particle composition and is a basic input parameter for carrying out relevant numerical simulation research. The international soil science community has developed a standardized soil texture classification system, of which the most widely adopted is the classification system proposed by the International Society of Soil Science (ISSS) [26]. The soil texture classification system proposed by ISSS is shown in Figure 5. As shown in the triangular diagram, three types of soil particles are classified according to the effective diameter, i.e., clay particles (<0.002 mm), powder particles (0.002∼0.02 mm), and sand particles (0.02∼2 mm). According to the percentage of the three types of soil particles, soil is classified into 12 types of soil texture categories, which are heavy clay, sandy clay, light clay, silty clay, sandy clay loam, loam, clay loam, silty clay loam, loamy sand, sand, sandy loam, and silt loam [26]. One soil texture from the 12 types of soil textures is randomly selected for each soil layer in the GPR model, and then five different soil particle proportions in the range corresponding to the texture category are randomly determined, which are represented in Figure 5 by five points. That is, the same layer of soil in the GPR model has the same soil texture category, but the specific proportion of internal soil in each layer can be different. This is to make the soil in the GPR model consistent with the soil condition of real trees since the soil around the roots is not only layered but also heterogeneous within the layers.

After determining the texture, particle proportion, temperature and moisture of each layer of soil in the GPR model presented in Figure 4, the electrical parameters of the soil can be calculated based on the experiential values of the specific gravity of the soil particles and the soil bulk density. These electrical parameters include the permittivity and conductivity of soil, which are important inputs of GPR simulations. The permittivity and conductivity of soil are calculated according to the Peplinski soil mixing model [27] as follows [28]:(1)$$\epsilon =\epsilon^{\prime }-j\epsilon^{\prime \prime }$$(2)$$\sigma =2\pi \epsilon_{0}f\epsilon^{\prime \prime }=0.05563f_{\mathrm{GHz}}\epsilon^{\prime \prime }$$(3)$$\epsilon_{\left(1.4∼18\;\mathrm{GHz}\right)}^{\prime }={\left(1+\frac{\rho_{b}}{\rho_{s}}\left(\epsilon_{s}^{a}-1\right)+f_{w}^{\beta^{\prime }}{\epsilon_{w}^{\prime }}^{a}-f_{w}\right)}^{1/a}$$(4)$$\epsilon_{\left(0.3∼1.3\;\mathrm{GHz}\right)}^{\prime }=1.15\epsilon_{\left(1.4∼18\;\mathrm{GHz}\right)}^{\prime }-0.68$$(5)$$\epsilon^{\prime \prime }=f_{w}^{\frac{\beta^{\prime \prime }}{a}}\left(\epsilon_{w}^{\prime \prime }+\frac{\sigma_{f}}{\omega \epsilon_{0}}\frac{\left(\rho_{s}-\rho_{b}\right)}{\rho_{s}f_{w}}\right)$$(6)$$\epsilon_{s}={\left(1.01+0.44\rho_{s}\right)}^{2}-0.062$$(7)$$\beta^{\prime }=1.2748-0.519\cdot S-0.152\cdot C$$(8)$$\beta^{\prime \prime }=1.33797-0.603\cdot S-0.166\cdot C$$(9)$$\epsilon_{w}=\epsilon_{w,\infty }+\frac{\epsilon_{w,s}-\epsilon_{w,\infty }}{1+j\omega t_{0,w}}$$(10)$$\sigma_{f}=0.0467+0.2204\rho_{b}-0.411S+0.6614C$$

Equation (Equation 1) is the complex representation of the relative permittivity ϵ of soil. $\epsilon^{\prime }$ is the real part, $\epsilon^{\prime \prime }$ is the imaginary part, and j is the imaginary unit. Equation (Equation 2) is a conversion formula for conductivity σ of soil and $\epsilon^{\prime \prime }$, where *f* represents the frequency in Hz, $f_{\mathrm{GHz}}$ represents frequency in GHz, and $\epsilon_{0}$ is the vacuum dielectric constant. Equation (Equation 3) is the formula for the real part of the relative permittivity from 1.4∼18 GHz, where $\rho_{b}$ is the soil bulk density, $\rho_{s}$ is the specific gravity of the soil particles, with a constant a=0.65, and $f_{w}$ is the volume fraction of water. When the frequency is 0.3∼1.3 GHz, a linear correction is made by Equation (Equation 4). In Equation (Equation 5), $\epsilon_{s}$ is the relative permittivity of sand particles, $\epsilon_{w}$, $\epsilon_{w}^{\prime }$, $\epsilon_{w}^{\prime \prime }$ are the relative permittivity of water and its real and imaginary parts, respectively, and ω is the angular frequency. The $\beta^{\prime }$, $\beta^{\prime \prime }$ in Equations (Equation 3) and (Equation 5) are given in Equations (Equation 7) and (Equation 8), where S and C are the percentages of sand and clay particles. In Equation (Equation 9), $\epsilon_{w,s}$ and $\epsilon_{w,\infty }$ are the relative permittivity of water at 0 and infinite frequencies, respectively, and $t_{0,w}$ is the relaxation time of the water, which is a variable that depends on the temperature of the water, and is about 9.23 picoseconds at 20 °C. Equation (Equation 10) is given in Equation (Equation 5), as the intermediate variable $\sigma_{f}$ takes values at frequencies of 0.3∼1.3 GHz.

The circular, semicircular, square, and triangular objects are randomly selected and rotated at any angle from 0 to 360 degrees before being embedded in the second layer of the GPR simulation model, with a depth ranging from 20 cm to 30 cm. The radii of the circle and semicircle, as well as the radius of the circumcircle of the triangle, range from 5 to 10 cm, while the length of the rectangle is between 10 and 15 cm, and the width is between 4 and 6 cm. The relative permittivity of these buried objects is randomly assigned in the range of 2 to 32, and this operation is to achieve the purpose of simulating the distribution of underground tree root systems.

### 3.2. Simulation Data Preprocessing

In order to improve the discernibility of reflected waves in GPR B-scan data generated by gprMax, three operations of background removal, gain, and normalization are applied to the original B-scan data from gprMax in sequence, which are calculated as follows:(11)$${x^{\prime }}_{ij}=x_{ij}-\frac{1}{N}\sum_{j=1}^{N}x_{ij}$$(12)$${x^{\prime \prime }}_{ij}={x^{\prime }}_{ij}\cdot {\left(e^{\left(2\times 10^{7}\cdot i\cdot \Delta t\right)}\right)}^{5}$$(13)$${x^{\prime \prime \prime }}_{ij}=\frac{{x^{\prime \prime }}_{ij}-\min\left({x^{\prime \prime }}_{ij}\right)}{\max\left({x^{\prime \prime }}_{ij}\right)-\min\left({x^{\prime \prime }}_{ij}\right)}$$

Equation (Equation 11) removes the direct wave interference from the echo signal, where *i* is the time step, *j* is the data channel number, $x_{ij}$ is the original GPR B-scan data, *N* is the total number of channels, and $x_{ij}^{\prime }$ is the result after background removal [29]. As GPR detects the subsurface, the electromagnetic signal propagates downward and gradually decays, resulting in a very small amplitude of deep echo signals. To ensure that the echo signals of different depths can be displayed, it is necessary to compensate for these rapidly decaying deep signals in the GPR B-scan data. This process is called time gain [30], which is designed to make up for the loss of energy generated by the radar wave re-propagation process, thus enhancing the amplitude of the echo waves in the deep layers. Correct adjustment of the gain can enhance the recognition ability of GPR on underground targets, and reduce the risk of misjudgment. The time gain is calculated as Equation (Equation 12), and the sampling interval of the signal is $\Delta t=5.89664\times 10^{-12}s$. $x_{ij}^{\prime \prime }$ is the gain-corrected signal [30]. Equation (Equation 13) is the operation of normalizing the data, $x_{ij}^{\prime \prime \prime }$ is the normalized GPR B-scan data, and min and max denote the minimum and maximum values in the GPR B-scan data, respectively. After completing the above operations, the GPR B-scan image is scaled to a size of 128×128 using the nearest neighbor interpolation method [31], which is easy to input by the following deep learning models. The permittivity distribution map corresponding to the GPR B-scan is also normalized and scaled to 128×128 size as the first training label for the subsequent deep learning model, and the second training label is obtained by extracting the edges of the preprocessed permittivity distribution map, labeling the edge value as 1 and the other parts as 0.

### 3.3. Comparison with Other Methods

#### 3.3.1. Loss Function and Evaluation Indicators

In the experiments of this paper, the mean square error (MSE) [32] equation is used as a loss function for the proposed GPR inversion model. The loss function is defined as follows:(14)$${Loss}_{mse}=\frac{1}{B}\underset{b=1}{\sum^{B}}\;{\left(Y_{b}-{\hat{Y}}_{b}\right)}^{2}$$(15)$${Loss}_{\mathrm{total}}=\alpha \cdot Loss1_{mse}+\beta \cdot Loss2_{mse}$$
where ${Loss}_{\mathrm{total}}$ denotes the final loss function of the inversion model PyViTENet, $Loss1_{mse}$ is the loss function of the main inversion task, and $Loss2_{mse}$ is the loss function of the edge inversion auxiliary task. α and β denote the weights of the two loss functions, which are set to 1 and 0.01, respectively, according to the difference in the training difficulty of the two tasks. B is the batch size, and $Y_{b}$ and ${\hat{Y}}_{b}$ are the true values and the predicted values of the relative permittivity of underground material, respectively.

To quantitatively evaluate the performance of different GPR inversion methods, the following evaluation indicators are used to measure the difference between the predicted value and the ground truth of the relative permittivity of the underground material:(16)$$SSIM\left(Y,\hat{Y}\right)=\frac{1}{B}\underset{b=1}{\sum^{B}}\;\frac{\left(2\mu_{Yb}\mu_{\hat{Y}b}+c_{1}\right)\left(2\sigma_{Yb,\hat{Y}b}+c_{2}\right)}{\left(\mu_{Yb}^{2}+\mu_{\hat{Y}b}^{2}+c_{1}\right)\left(\sigma_{Yb}^{2}+\sigma_{\hat{Y}b}^{2}+c_{2}\right)}$$(17)$$MSE\left(Y,\hat{Y}\right)=\frac{1}{B}\underset{b=1}{\sum^{B}}\;{\left(Y_{b}-{\hat{Y}}_{b}\right)}^{2}$$(18)$$MAE\left(Y,\hat{Y}\right)=\frac{1}{B}\underset{b=1}{\sum^{B}}\;\left|Y_{b}-{\hat{Y}}_{b}\right|$$
where Equation (Equation 16) is the structural similarity index measure (SSIM), which measures the difference between the predicted and the real values from the global view with the criteria of brightness, contrast, and structure. These criteria are closer to the human way of visually sensing the image, and have a better effect in practical application: μ is the mean value of the image, σ is the variance of the image, $c_{1}=0.01$ and $c_{2}=0.03$ are stationary constants, and when the value of SSIM is equal to 1, it is concluded that the two compared data are exactly the same [33]. Equations (Equation 17) and (Equation 18) are the MSE and Mean Absolute Error (MAE) [32], respectively, which are used to evaluate the numerical difference between the predicted value and the real value.

#### 3.3.2. Experimental Environment and Parameter Setting

In this study, all simulations are performed on an Intel(R) Xeon(R) CPU Gold 5318S (2.1 GHz) with 64 G of RAM and an NVIDIA A100 GPU with 80 G of video memory. Pytorch 1.10 is used as the deep learning framework with CUDA version 11.7, and Python 3.9 is used as the programming language in Jupyterlab 3.4.6 platform to complete the network structure construction and testing.

There are a total of 20,000 sets of simulation data, which are divided into a training set, validation set, and test set at a ratio of 8:1:1 in the experiments of this study. Among these 20,000 sets of data, 10,000 sets of data are simulation data of a single object buried in the layered heterogeneous soil, and the other 10,000 sets of data are the simulation data of two objects buried in the layered heterogeneous soil. For the simulation scene where two objects are buried underground, it contains 5000 sets of data of the two objects when overlapped and 5000 sets of data of the two objects when separated. The AdamW optimizer [34] with a weight decay coefficient of 0.01 is used in the training process. The initial learning rate of the optimizer is 0.0005, and the learning rate is dynamically adjusted by cosine annealing decay [35]. The batch size is 64, and the total number of training iterations for the GPR inversion models is 150.

#### 3.3.3. Inversion Results of Simulation Data

Figure 6 shows the inversion results of different methods on the simulation dataset, which are the relative permittivity distribution map of underground soil and objects. In the experiment, the traditional inversion method FWI is given a pre-estimated permittivity distribution of underground material as the initial value, which is the real data after Gaussian blurring [36]. The FWI method is not only time-consuming but also has lower inversion accuracy than other deep learning inversion methods. For the case of single or two objects buried in the complex soil model, the inversion results of U-Net, GPRInvNet, and DMRF-UNet are not accurate enough. U-Net cannot clearly identify the dividing line between the second and third layers of soil. The inversion results in the third and fourth columns of Figure 6 show that when two objects are buried in the soil, there are even missing objects in the U-Net inversion results. Similarly, GPRInvNet can only invert the dividing line between the first and second layers of soil and identify the approximate location of buried objects, but the inversion of the relative permittivity values of the objects is not accurate. Since DMRF-UNet is a two-stage network, it removes the waveform information of the background medium as noise, which makes it barely able to identify the soil stratification, and the inversion accuracy of the relative permittivity of objects is also worse than U-Net and GPRInvNet. In general, TransUNet can invert the shape and the relative permittivity of underground objects, as well as the layering of the soil. However, when TransUNet inverts scatterers that are small in size and have relative permittivity close to the surrounding soil, the inversion results from TransUNet become blurred or even missing objects, such as the triangular scatterers in the first and fourth columns of Figure 6. EDMFEBs can give better inversion results when facing the above problems, but its inversion of the contours of smaller scatterers is unclear as shown in the fourth column of Figure 6. Compared with other inversion methods, PyViTENet can more accurately invert the relative permittivity of underground materials and identify soil layering. To further compare the results of different methods, Figure 7 shows an enlarged view of the results of the above inversion methods on the detailed structure within the soil. The comparison in Figure 7 shows that with the assistance of the edge inversion auxiliary task, PyViTENet is able to identify the heterogeneous structural details of the soil more clearly than other methods.

Table 1 presents the evaluation indicators, prediction time, and the number of parameters of different deep learning-driven inversion methods. Compared with other methods, PyViTENet requires the most prediction time, and its number of parameters ranks second among all these methods. However, the three evaluation indicators of MSE, MAE, and SSIM of PyViTENet are the best compared to other inversion methods, which further verifies the effectiveness of PyViTENet.

### 3.4. Ablation Experiment

In order to verify the effectiveness of each module in PyViTENet, some modules are removed to conduct experiments separately under the premise that conditions such as the training method and training data are kept the same. The results of the ablation experiments for each model are shown in Table 2, and Figure 8 shows a visual comparison of the model predictions, where (f) is the proposed model, (a) is the ground truths, (b) is the model using PyConvFEB to replace ViTFEB and eliminate the edge inversion auxiliary task, (c) is the model eliminating the edge inversion auxiliary task, (d) and (e) are models using PyConvFEB to replace ViTFEB and adopting different values of α and β in the edge inversion auxiliary task loss function, respectively, and (f) is structurally consistent with the proposed model (g) but uses different values of α and β.

Unsurprisingly, (b) has the worst prediction results. (c) has a small decrease in accuracy compared to (g) due to the elimination of the edge inversion auxiliary task, and its predictions are less fine-grained in edge detail than (g). (c) and (e) have a significant decrease in accuracy, indicating that ViTFEB brings a stronger characterization capability to the model, which plays an important role in improving the prediction accuracy. Notably, (d) and (e) as well as (f) and (g) adjust the values of α and β with the same model structure. It can be found that when β is closer to α, the auxiliary task dominates the training of the model, leading to the poorer performance of the main task. Therefore, a smaller β is used here to obtain the best model (g).

## 4. Experiments and Analysis of Real Data

### 4.1. Real Dataset and Experimental Implementation

The inversion experiments of the measured data are carried out in the laboratory provided by Xinjiang Agricultural University. The measurements are conducted using a GER-10 GPR system in an indoor experimental box with a wave-absorbing material. The GPR system with a GER900A antenna is used to collect B-scans in experiments. The center frequency of the antenna is 900 MHz, the length of the antenna is 30 cm, and the distance between the transmitter and the receiver is 14 cm. The GPR utilizes a bow-tie antenna. The equipments are sourced from Zhongdian Zhongyi Intelligent Technology Development Co., Ltd., Qingdao, China.

As shown in Figure 9, the soil in the experimental box is dry silt loam from the Xishan area of Urumqi, Xinjiang, with a depth of 0.6 m, a length of 1 m, and a width of 0.8 m. The buried objects in the experimental box are three kinds of wooden objects: two cylindrical wood blocks with diameters of 6 and 7 cm, and lengths of 15 cm and 10 cm, respectively, made of wetted fir, and a rectangular wood board of 18 cm × 18 cm × 3 cm, made of basswood. The relative permittivity of the soil is 4 [37], the dry wood is 5, and the wetted wood is 10 [38]. For each measurement, one object is selected and buried at three depths of 10 cm, 15 cm, and 20 cm. The length of the measurement line is 0.7 m, with a total of 50 traces of data, each with 512 time sampling points and a sampling time window of 10 ns. A total of 190 B-Scan data are collected, including 70 sets of data for the cylindrical wood block, and 120 sets of data for the wetted and dried wood boards, respectively. The wood object is placed in two ways: horizontally and vertically. The collected data are divided into training and testing sets in a ratio of 9:1.

### 4.2. Analysis of the Experimental Results

Due to the difference between the simulation experiment in Section 3.3.3 and the real measurement scenario, 2000 sets of simulation data that are consistent with the soil and buried material parameters of the experimental site need to be added for pre-training before the inversion of the real measurement data. These new simulation data are used as initial parameters for the training of the inversion model for the real measured data [39]. The pre-training loads the models weights obtained from the simulation experiments in Section 3.3.3.

After the first step of the simulation experiment, the information from 20,000 sets of simulation data is already included in the pre-loaded weights. In the second step, after loading the weights from the first step, transitional training is conducted using 2000 sets of simulation data, as they are closer to the real measured data than the data from the first step. In the third step, the weights obtained from the previous two steps are pre-loaded into the deep learning model, followed by training on the real measured data. There is a distribution discrepancy between the real data and simulation data. Directly using all 20,000 sets of simulation data may cause the model to overfit idealized scenarios, reducing its generalization ability to real data. By selecting 2000 sets of simulation data that are highly consistent with the experimental site parameters, the domain gap can be reduced, thereby improving the transfer learning performance.

Table 3 demonstrates the evaluation metrics of each deep learning method. Among them, the accuracy of U-Net, DMRF-UNet, and GPRInvNet is much lower than that of the other models. Due to their poor inversion on the simulation dataset, they do not successfully perform inversion on the real data by transfer learning. Figure 10 illustrates the prediction results for some of the test set data, where the buried objects in columns 1 and 2 are cylindrical wooden blocks, and those in columns 3, 4, and 5 are rectangular wooden boards. The wooden boards in column 4 are drier, and the other objects are wetted. From the results, it is obvious that FWI can only predict the approximate position of the scatterer but fails to accurately portray the shape and permittivity information of the object. U-Net, GPRInvNet, and DMRF-UNet are hardly able to retrieve any valid information for the measured data. TransUNet has a lot of background noise in the inversion results, causing the scatterers to be less clearly visible. Compared with other methods, EDMFEBs and PyViTENet provide a more accurate inversion performance of the shape, position and relative permittivity for wetted objects, while the inversion results of PyViTENet are more accurate for the boards placed vertically in column 3, proving the generalizability of the proposed model to different states of buried objects. However, since the board in column 4 is dry and the permittivity of the board is very close to that of the soil, the inversion of all the methods is not satisfactory. The GPR inversion relies on the contrast in permittivity. Since the permittivity of dry wood and dry soil is relatively similar, the reflection intensity of radar waves is reduced, making it difficult for GPR to clearly distinguish wooden targets from the surrounding soil. This may result in weak or even invisible target echoes. Due to the weak reflected signal, the target boundaries become indistinct, making it challenging to accurately reconstruct its shape.

## 5. Conclusions and Perspectives

In this paper, a deep learning-based ground-penetrating radar inversion model PyViTENet is proposed and validated, aiming to improve the accuracy of non-destructive detection of tree roots in soil. The model improves the representation ability of the model by fusing Pyramid Convolution PyConv and Vision Transformer. Meanwhile, the task for the inversion of the boundary of the dielectric constant distribution map is introduced so that the model pays more attention to the part of small structural changes inside the soil and the boundary information between the object and the soil, achieving the efficient extraction of complex subsurface structural features. The model is trained by constructing a simulation dataset containing heterogeneous stratified soils with different humidity, temperature, and texture, as well as buried objects simulating the root structure of a tree. The model trained on the simulation data is eventually transferred to the inversion problem for the real data.

The experimental results show that the PyViTENet model outperforms other existing models on the simulated dataset, and enables a more accurate inversion of the scatterer’s permittivity and soil stratification. By transfer learning on real measurement data, the model demonstrates good generalization ability and is successfully applied to buried material detection under real soil conditions, providing a basis for research on root detection in complex soils. Although high inversion accuracy has been achieved on laboratory measurement data, the inversion of more complex nonhomogeneous soil internal properties in the field still needs further research. In future work, the inversion of real measured data for buried material in nonhomogeneous layered soils will be further studied.

## Figures and Tables

**Figure 1 sensors-25-00947-f001:**
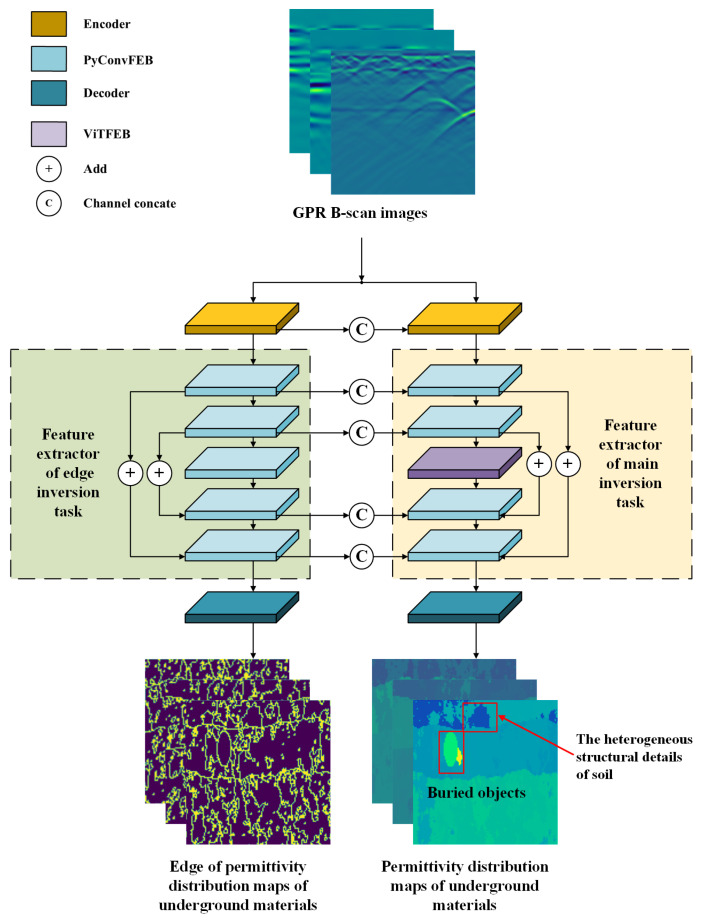
The structure of the GPR inversion model PyViTENet.

**Figure 2 sensors-25-00947-f002:**
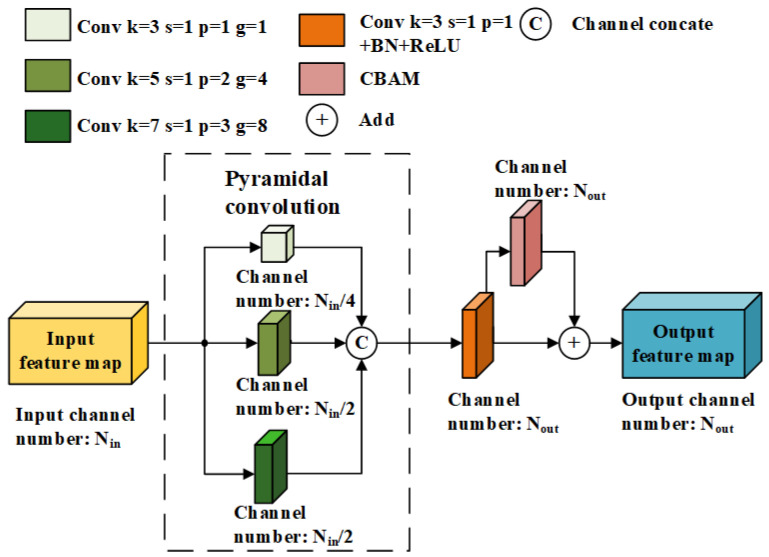
The structure of PyConvFEB.

**Figure 3 sensors-25-00947-f003:**
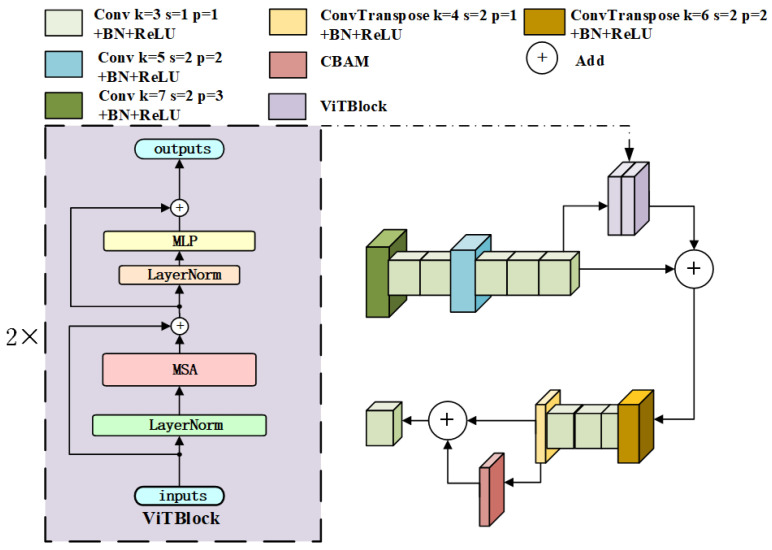
The structure of the ViTFEB module.

**Figure 4 sensors-25-00947-f004:**
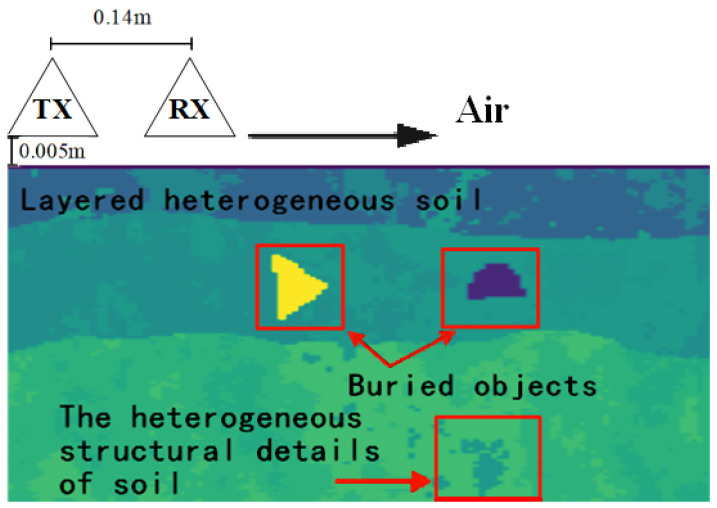
GPR model.

**Figure 5 sensors-25-00947-f005:**
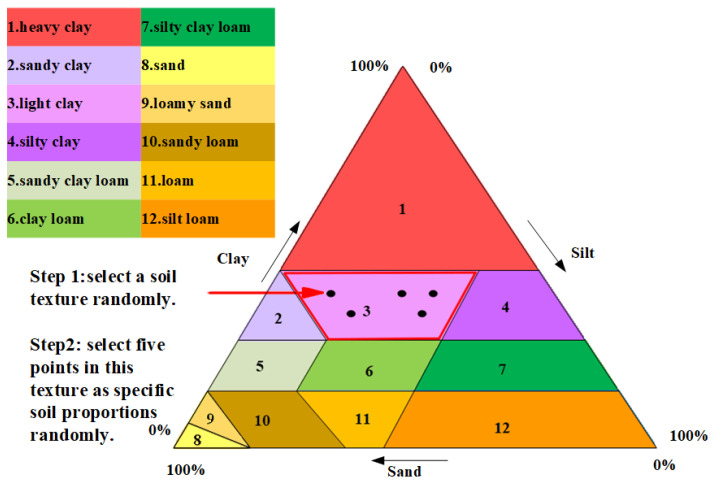
Soil texture classification.

**Figure 6 sensors-25-00947-f006:**
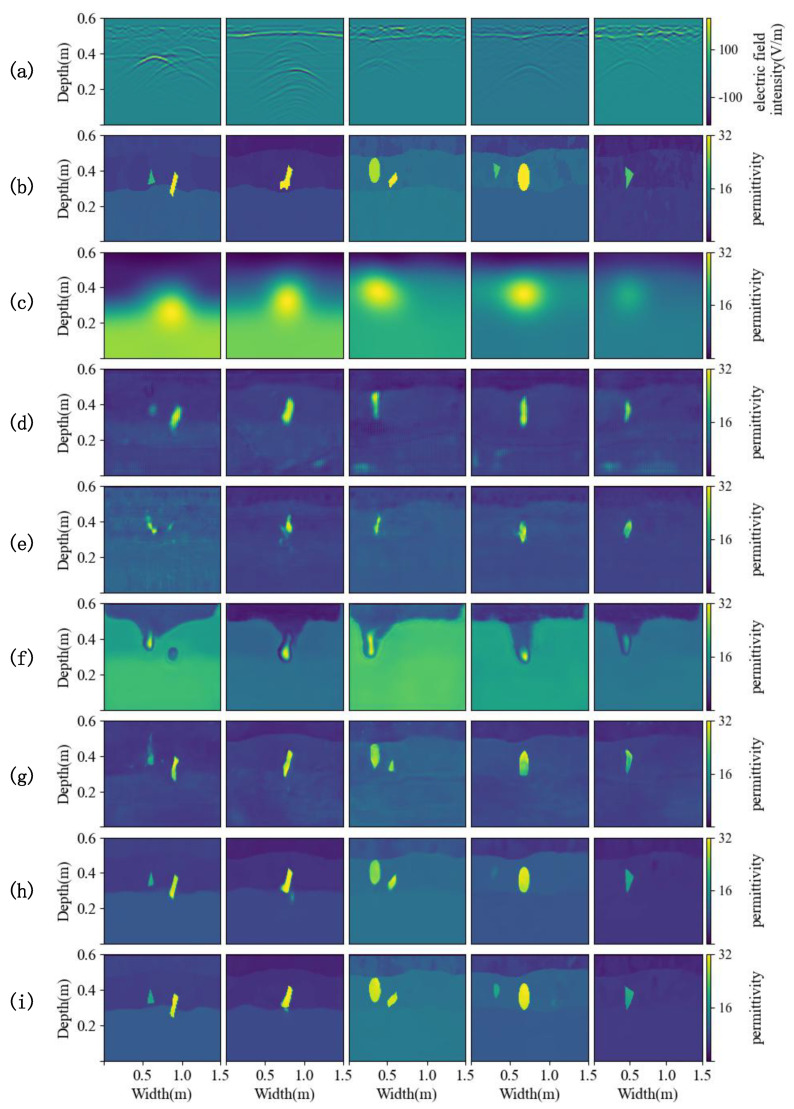
Inversion results. (**a**) Preprocessed B-scans. (**b**) Ground truths of the relative permittivity distribution of underground material. (**c**–**i**) Predicted relative permittivity distributions from (**c**) FWI, (**d**) U-Net, (**e**) GPRInvNet, (**f**) DMRF-UNet, (**g**) TransUNet, (**h**) EDMFEBs and (**i**) PyViTENet.

**Figure 7 sensors-25-00947-f007:**
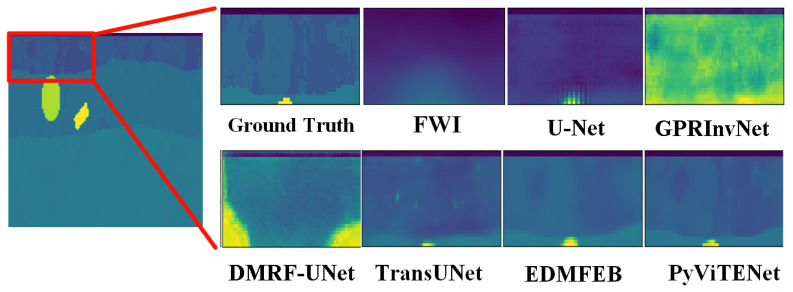
An enlarged view of the inversion results on the detailed structure within the soil obtained by different inversion methods. The right side of the figure is a magnified view of the inversion results corresponding to the red area on the left side. The different colors represent the same values as in Figure 6.

**Figure 8 sensors-25-00947-f008:**
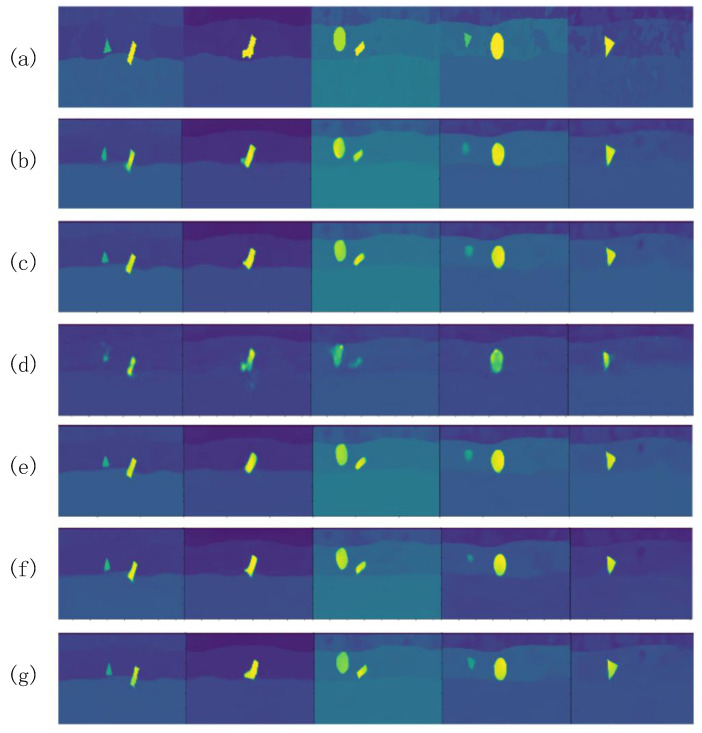
Results of ablation experiment. (**a**) Ground truths. (**b**) The main inversion task using PyConvFEBs only. (**c**) The main inversion task using PyConvFEBs and ViTFEB. (**d**) The main inversion task using PyConvFEBs and the edge inversion auxiliary task (α=1, β=1). (**e**) The main inversion task using PyConvFEBs and the edge inversion auxiliary task (α=1, β=0.01). (**f**) PyViTENet (α=1, β=1). (**g**) PyViTENet (α=1, β=0.01). The different colors represent the same values as in Figure 6.

**Figure 9 sensors-25-00947-f009:**
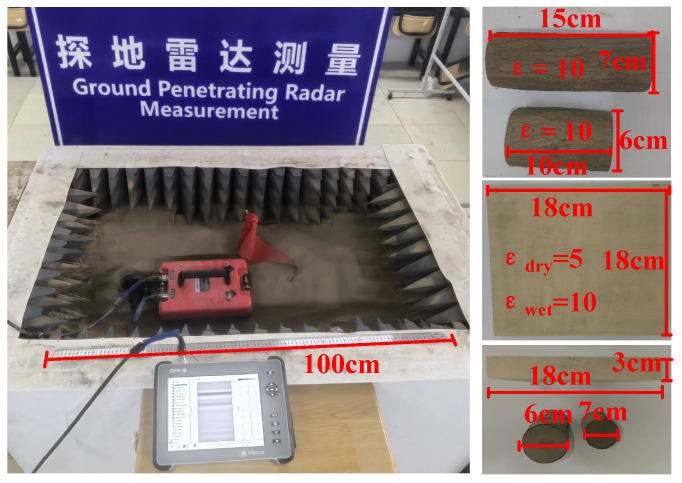
The experimental site of the real dataset.

**Figure 10 sensors-25-00947-f010:**
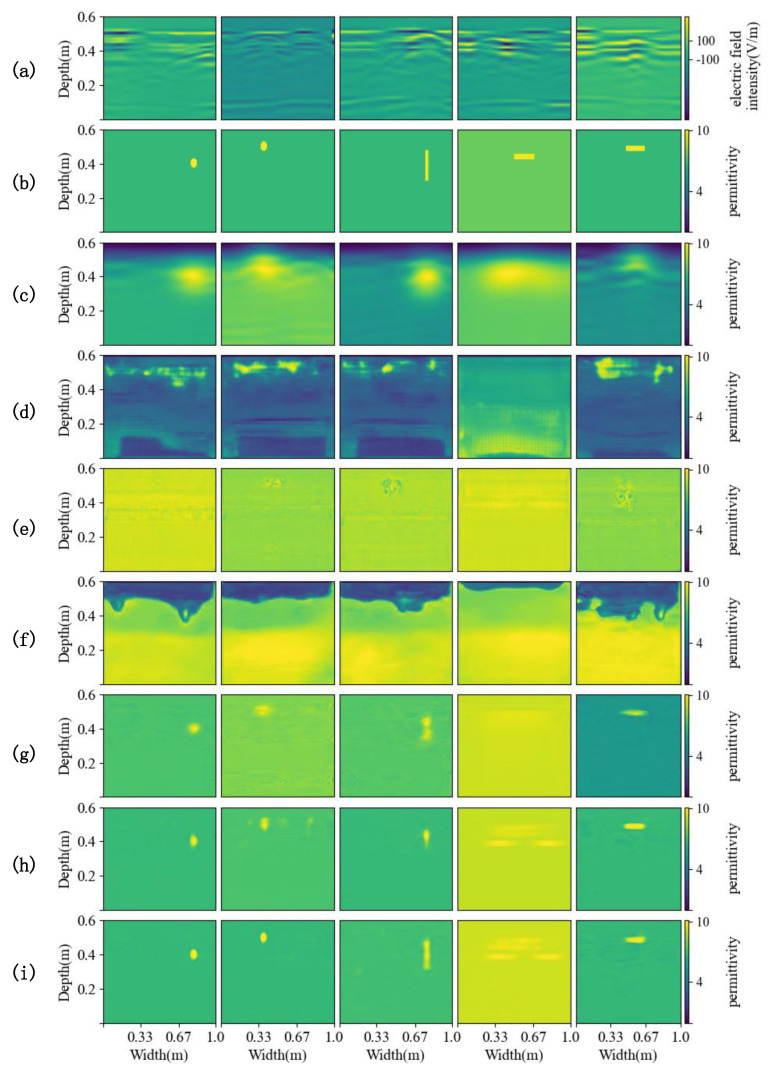
Comparison of the inversion results of each method on the real dataset. (**a**) Preprocessed B-scans. (**b**) Ground truths of the relative permittivity distribution of underground material. (**c**–**i**) Predicted relative permittivity distributions from (**c**) FWI, (**d**) U-Net, (**e**) GPRInvNet, (**f**) DMRF-UNet, (**g**) TransUNet, (**h**) EDMFEBs and (**i**) PyViTENet.

**Table 1 sensors-25-00947-t001:** Comparison of evaluation indicators, prediction time, and the number of parameters of different inversion methods.

Model	MSE ($10^{-4}$) ↑	MAE ($10^{-2}$) ↑	SSIM ↓	Prediction Time(s)	Number of Parameters ($10^{6}$)
PyViTENet	5.18	2.2760	0.9453795	1.14	18.9
EDMFEBs	5.74	2.3958	0.9364224	0.88	7.9
U-Net	9.99	3.1612	0.9158119	0.83	1.9
TransUNet	15.98	3.9975	0.8852904	1.12	66.8
GPRInvNet	21.93	4.6829	0.8367578	0.85	1.4
DMRF-UNet	47.16	6.8672	0.5886880	0.91	7.7

The up arrow (↑) represents ascending order. The down arrow (↓) represents descending order.

**Table 2 sensors-25-00947-t002:** Ablation experiment of PyViTENet.

Model	MSE (10^−4^)	MAE (10^−2^)	SSIM
PyConvFEBs	5.85	2.4195	0.9408528
PyConvFEBs + ViTFEB	5.54	2.3537	0.9442102
PyConvFEBs + Edge Task (α = 1, β = 1)	8.74	2.9563	0.9120258
PyConvFEBs + Edge Task (α = 1, β = 0.01)	5.70	2.4021	0.9422152
PyViTENet (α = 1, β = 1)	8.36	2.8714	0.9361667
PyViTENet (α = 1, β = 0.01)	5.18	2.2760	0.9453795

**Table 3 sensors-25-00947-t003:** Comparison of evaluation indicators of different inversion methods on the real dataset.

Model	MSE (10−4) ↑	MAE (10−2) ↑	SSIM ↓
PyViTENet	3.31	1.1618	0.9715505
EDMFEBs	3.45	1.6213	0.9715072
TransUNet	93.40	9.64	0.7297934
U-Net	297.42	15.1606	0.6747670
GPRInvNet	1260.75	30.2256	0.5292989
DMRF-UNet	2416.36	48.4334	0.4273940

The up arrow (↑) represents ascending order. The down arrow (↓) represents descending order.

## Data Availability

The raw data supporting the conclusions of this article will be made available by the authors on request.

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
