# Peer review of "Deep Learning-Based Ground-Penetrating Radar Inversion for Tree Roots in Heterogeneous Soil"

_sensors, 2025, doi:10.3390/s25030947_

Round 1
Reviewer 1 Report
Comments and Suggestions for Authors
I much appreciated the contribution. It appears to be neatly written (a few typos, maybe? and some need to break, whether possible, quite long paragraphs into several ones, one idea at a time, as one says, to facilitate the reading, especially the Introduction).
It is quite complete in scientific/technical terms, and well-discussed material, original as well. Here, I have some comments, however. I
In section 3.1.1, simulations are led with care indeed, yet nothing is said on the antennas that are hypothesized, noticing that Gprmax appears to be able to model antennas to an extent. In the drawing, it looks like horn antennas ... is that the case? And does one speak on S parameters, or just consider that one somehow collects the pulse scattered back from the soil w. objects in it? It would be useful to tell us something about the above, especially if the aim is to get pertinent sets of data effectively collected by antennas (even if somewhat idealized still) for training the AI system, pertinent being vs. the experiments thereafter, I mean.
In a sense, this is about the same later on, one speaks about objects, but they pretty much remain unspecified for what concerns their precise geometries and also, to a lesser extent their depth of embedding. True, they are images of ground truth, yet to be more specific on dimensions should help.
One remark, the ablation experiment, is it necessary or even useful? What do the authors wish to prove here? I might have missed the point, I agree, but others might also. Maybe, it is just about the way things are commented upon? I do not sugegst to take that material out, however.
Last, about the real data. I wonder why not using real tree roots (properly excavated), and if one cannot excavate, something with as an example branching as usual with roots and made in wood, dry or wetted appropriately, instead of the mentioned blocks? Not sure by the way that the rectangular plate fits so well the purpose of the contribution, as it is insisted on tree roots. I agree this is just a start, however, so I do not suggest change, and I do appreciate the experiments. Obviously also, as indeed underlined, the experimental data do not suffice to train and test, and it is of utmost importance to mix them with simulated data, the authors do that, it seems properly, even if I got a bit fuzzy on the fact that one had 20.000 at the start and now 2.000 only, that is, why one did not run with the 20.000 ...
That is, I believe that the notion of consistency at the top of section 4.2 should be made more explicit, or did I just alas miss something? In brief, I would suggest to expand a bit the top paragraph of 4.2, since just mentioning deep transfer learning as set up in [35] appears to leave aside much of what the authors do in practice.
To conclude, it is of my knowledge that a multiauthored volume on Ground Penetrating Radar, From Theoretical Endeavors to Computational Electromagnetics Signal, Processing, Antenna Design, and Field Applications, has appeared w. ISTE-Wiley in Spring 2024, any interest in it? A lot of references appear to be put to the fore within this volume, including tree root characterization, and evidently GPR design and use at large, which might have a good linkage to the present investigation.
Author Response
Comments 1: I much appreciated the contribution. It appears to be neatly written (a few typos, maybe? and some need to break, whether possible, quite long paragraphs into several ones, one idea at a time, as one says, to facilitate the reading, especially the Introduction.
Response 1:Thank you for your valuable feedback. We appreciate your positive comments on our work. We have carefully reviewed the manuscript and corrected any typos.Additionally, we have revised long paragraphs, especially in the Introduction, by breaking them into shorter paragraphs to enhance readability.
Comments 2: In section 3.1.1, simulations are led with care indeed, yet nothing is said on the antennas that are hypothesized, noticing that Gprmax appears to be able to model antennas to an extent. In the drawing, it looks like horn antennas ... is that the case? And does one speak on S parameters, or just consider that one somehow collects the pulse scattered back from the soil w. objects in it? It would be useful to tell us something about the above, especially if the aim is to get pertinent sets of data effectively collected by antennas (even if somewhat idealized still) for training the AI system, pertinent being vs. the experiments thereafter, I mean.
Response 2:Thank you for your insightful comments. We acknowledge the importance of specifying the antenna model used in the simulations. In our study, we utilized the built-in antenna model provided by gprMax, which is based on a resistively loaded bow-tie antenna rather than a horn antenna. We have now clarified this in Section 3.1.1.The added sentence is as follows:
The type of antenna design used by default in gprMax is the bow-tie antenna.(Line.210)
The GPR utilizes a bow-tie antenna.(Line.430)
Regarding the signal processing approach, we do not explicitly consider S-parameters. Instead, we focus on collecting the B-scan from the soil and embedded objects, as is commonly done in GPR data acquisition. The main objective is to generate simulation data that align well with experimental measurements, ensuring consistency for training the deep learning model.
Comments 3: In a sense, this is about the same later on, one speaks about objects, but they pretty much remain unspecified for what concerns their precise geometries and also, to a lesser extent their depth of embedding. True, they are images of ground truth, yet to be more specific on dimensions should help.
Response 3:The additional details have been provided as follows:
The circular, semicircular, square, and triangular objects are randomly selected and rotated at any angle from 0 to 360 degrees before being embedded in the second layer of the GPR simulation model, with a depth ranging from 20 cm to 30 cm.The radius of the circle and semicircle, as well as the radius of the circumcircle of the triangle, range from 5 to 10 cm, while the length of the rectangle is between 10 and 15 cm, and the width is between 4 and 6 cm.(Line 289)
Comments 4: One remark, the ablation experiment, is it necessary or even useful? What do the authors wish to prove here? I might have missed the point, I agree, but others might also. Maybe, it is just about the way things are commented upon? I do not sugegst to take that material out, however.
Response 4:The ablation experiment is crucial for demonstrating the effectiveness of each individual component of the PyViTENet model. By systematically removing or modifying certain modules, we aim to highlight how each component contributes to the overall performance and help to understand the interactions between different parts of the model. The goal is not only to show the superior performance of the complete model but also to identify which modules or features play the most significant role in enhancing the model's prediction accuracy and generalization. We believe this analysis provides transparency and insights into how the design choices influence the final results, which could be useful for future work and model improvements.
Comments 5: Last, about the real data. I wonder why not using real tree roots (properly excavated), and if one cannot excavate, something with as an example branching as usual with roots and made in wood, dry or wetted appropriately, instead of the mentioned blocks? Not sure by the way that the rectangular plate fits so well the purpose of the contribution, as it is insisted on tree roots. I agree this is just a start, however, so I do not suggest change, and I do appreciate the experiments. Obviously also, as indeed underlined, the experimental data do not suffice to train and test, and it is of utmost importance to mix them with simulated data, the authors do that, it seems properly, even if I got a bit fuzzy on the fact that one had 20.000 at the start and now 2.000 only, that is, why one did not run with the 20.000 ...
Response 5:Due to the difficulty in creating training labels, specifically the dielectric constant distribution maps, for irregular tree roots, we have currently used scatterers with regular geometries. In future research, we aim to develop more complex and realistic datasets that better reflect actual tree root structures.
There is a distribution discrepancy between real and simulated data. Directly using all 20,000 simulated data points could lead the model to overfit to an idealized scenario, reducing its ability to generalize to real data. By selecting 2,000 simulated data points that closely match the experimental field parameters, the domain gap can be narrowed, thereby improving the transferability of the model.
Comments 6 That is, I believe that the notion of consistency at the top of section 4.2 should be made more explicit, or did I just alas miss something? In brief, I would suggest to expand a bit the top paragraph of 4.2, since just mentioning deep transfer learning as set up in [35] appears to leave aside much of what the authors do in practice.
Response 6:The additional details abort transfer learning have been provided as follows:
After the first step of the simulation experiment, the information from 20,000 sets of simulation data is already included in the pre-loaded weights. In the second step, after loading the weights from the first step, transitional training is conducted using 2,000 sets of simulation data, as they are closer to the real measured data than the data from the first step. In the third step, the weights obtained from the previous two steps are pre-loaded into the deep learning model, followed by training on the real measured data. There is a distribution discrepancy between real data and simulation data. Directly using all 20,000 sets of simulation data may cause the model to overfit idealized scenarios, reducing its generalization ability to real data. By selecting 2,000 sets of simulation data that are highly consistent with the experimental site parameters, the domain gap can be reduced, thereby improving the transfer learning performance.(Line.452)
Comments 7:To conclude, it is of my knowledge that a multiauthored volume on Ground Penetrating Radar, From Theoretical Endeavors to Computational Electromagnetics Signal, Processing, Antenna Design, and Field Applications, has appeared w. ISTE-Wiley in Spring 2024, any interest in it? A lot of references appear to be put to the fore within this volume, including tree root characterization, and evidently GPR design and use at large, which might have a good linkage to the present investigation.
Response 7:Thank you for your suggestion. This book has been very helpful for our future research and has already been added to the reference list.

Reviewer 2 Report
Comments and Suggestions for Authors
The presented manuscript relates to an important area of using machine learning to process ground penetrating radar data. The authors propose an inversion method for detecting roots in heterogeneous soils. But the problem with this the task is that we do not know at what angle the GPR line will intersect the root. Because of this, we can receive a very wide range of patterns. The authors tried to prove the efficiency of their method in very simple models that cannot be correlated with real root systems. Therefore, it is not entirely correct to declare in the title that the proposed inversion approach for detecting tree roots. Rather, these are just some locally buried objects. It is also not very clear whether the features of heterogeneous soils will be detected. Some success was shown on synthetic models, but how it works on real objects is not yet clear. From the data got, machine learning-based algorithms are best at detecting local objects that have a clear signature on GPR records as a diffracted wave. The inversion of a more complex organization of the subsurface environment requires further research. If possible, please post your thoughts on how to solve similar problems in the discussion section. But I support such research, since they are very promising to develop the GPR method.
The following comments can be highlighted:
1. How can the proposed algorithm consider the non-stationary effect associated, for example, with the migration of the GPR signal, anisotropy of properties, etc.?
2. Please show its electro physical parameters, when you describing the model in section 3.1.1. Why do you use only the relative permittivity indicator, ignoring the electrical conductivity, although it determines the attenuation of the signal and reflection?
3. Line 120. I think, this is not quite true, the inversion algorithms are still in the development stage, while high-speed analysis and allocating GPR interfaces, patterns, etc. are used.
3. Line 252-281. Why did you choose the Peplinski model and not, for example, CRIM? This formula allows you to predict the real and imaginary parts of the permittivity, but how did you calculate the complex relative permittivity? Why did you decide this formula would give correct estimates for the entire range of soil grain size composition, if the authors tested it mainly on clay soils? And please provide a link to the original source Peplinski et al. (1995).
4. Line 421. Where did you get these values?
5. Figure 10. Why do the model indicators (b) not correspond to the specified parameters of the dielectric constant? Also, different ranges are specified for all the figures, which does not allow us to assess the accuracy of the selection.
6. Figure 10. The resulting radargrams show two interfaces at a depth of 0.4 and 0.1 from the bottom of the cell. Please explain why the algorithms did not reflect their existence?
7. Your data (Fig. 10 column 4) indicate that some level of contrast for the electrical properties is required for the inversion to work correctly. Please add some additional explanations to this.
Author Response
Comments 1: How can the proposed algorithm consider the non-stationary effect associated, for example, with the migration of the GPR signal, anisotropy of properties, etc.?
Response 1: We sincerely appreciate the reviewer’s insightful question regarding the consideration of non-stationary effects, such as the migration of the GPR signal and the anisotropy of properties, in our proposed algorithm.Deep learning models, when trained on sufficiently a large number of datasets, have the capability to implicitly learn complex physical relationships, including those associated with wave propagation and material anisotropy. This reduces the need for extensive manual preprocessing, as the network can capture these effects directly from the data. However, we acknowledge that incorporating prior physical knowledge into the preprocessing stage could further enhance the efficiency and accuracy of the model, particularly by reducing the required training data and mitigating potential errors.In future work, we plan to integrate physical constraints and domain-specific priors into our data processing pipeline. This approach would not only improve the generalization capability of the model but also reduce dependency on large training datasets, making the method more practical for real-world applications.
Comments 2: Please show its electro physical parameters, when you describing the model in section 3.1.1. Why do you use only the relative permittivity indicator, ignoring the electrical conductivity, although it determines the attenuation of the signal and reflection?
Response 2: Here is the supplement sentence in section 3.1.1.:
The permittivity distribution maps are chosen as the training label for the dataset instead of the conductivity distribution maps.The permittivity is directly related to the propagation time and reflection intensity of electromagnetic waves, both of which can be extracted from GPR data with high accuracy. This makes it particularly suitable for detailed soil property inversion. In contrast, Conductivity primarily influences signal attenuation, which tends to have lower resolution and is affected by multiple factors, such as noise, antenna characteristics, and varying environmental conditions. As a result, extracting precise conductivity distributions from B-scan data remains more difficult.(Line.222)
Comments 3: Line 120. I think, this is not quite true, the inversion algorithms are still in the development stage, while high-speed analysis and allocating GPR interfaces, patterns, etc. are used.
Response 3: Thank you for your correction. The inaccurate statement has been removed.
Comments 4: Line 252-281. Why did you choose the Peplinski model and not, for example, CRIM? This formula allows you to predict the real and imaginary parts of the permittivity, but how did you calculate the complex relative permittivity? Why did you decide this formula would give correct estimates for the entire range of soil grain size composition, if the authors tested it mainly on clay soils? And please provide a link to the original source Peplinski et al. (1995).
Response 4: We chose the Peplinski model primarily because it is directly supported in the simulation software gprMax, which provides built-in functions and code calculate the complex relative permittivity. This makes the simulation process more convenient and ensures consistency in modeling.Additionally, our simulation data include soils of different textures, including clay. Since the Peplinski model has been shown to perform better than the CRIM model for soils with clay content and is widely used in agricultural applications, we opted to use it in this study. However, we acknowledge the importance of evaluating different models, and in future research, we plan to explore the CRIM model as well.We have added a reference to the original work by Peplinski et al. (1995) in the revised manuscript.
Comments 5: Line 421. Where did you get these values?
Response 5: Due to the lack of equipment for accurately measuring permittivity, the permittivity of the object and soil is taken from references.Relevant references have been added.
Comments 6: Figure 10. Why do the model indicators (b) not correspond to the specified parameters of the dielectric constant? Also, different ranges are specified for all the figures, which does not allow us to assess the accuracy of the selection.
Response 6: We feel sorry for the incorrect labeling of the colorbars in Figure 10. which represents permittivity.The figure has been revised.
Comments 7: Figure 10. The resulting radargrams show two interfaces at a depth of 0.4 and 0.1 from the bottom of the cell. Please explain why the algorithms did not reflect their existence?
Response 7: The experiment of real dataset is relatively simpler compared to the simulation experiment, with train labels containing only the buried object and the assumed homogeneous soil.
Therefore, the inversion results primarily focus on the buried object and the average relative permittivity of the soil in the experiment of real dataset.These interfaces may reflect the actual heterogeneous structure present in the soil and the bottom of the experimental box.The algorithms will not reflect them.In future research, we will strive to establish more complex real datasets to address this issue.
Comments 8: Your data (Fig. 10 column 4) indicate that some level of contrast for the electrical properties is required for the inversion to work correctly. Please add some additional explanations to this.
Response 8: Thank you for your professional advice. Here is the supplement sentence:
GPR inversion relies on the contrast in permittivity. Since the permittivity of dry wood and dry soil is relatively similar, the reflection intensity of radar waves is reduced, making it difficult for GPR to clearly distinguish wooden targets from the surrounding soil. This may result in weak or even invisible target echoes. Due to the weak reflected signal, the target boundaries become indistinct, making it challenging to accurately reconstruct its shape.(Line.479)
